# Ursodeoxycholic acid and severe COVID-19 outcomes in a cohort study using the OpenSAFELY platform
Ruth E. Costello [1] ✉, Karen M. J. Waller[2,3,4], Rachel Smith[5], George F. Mells[5,6], Angel Y. S. Wong [1], Anna Schultze[1], Viyaasan Mahalingasivam[1], Emily Herrett [1], Bang Zheng[1], Liang-Yu Lin[1], Brian MacKenna[7], Amir Mehrkar [7], Sebastian C. J. Bacon[7], Ben Goldacre[7], Laurie A. Tomlinson [1], John Tazare[1,9], Christopher T. Rentsch [1,8,9], the OpenSAFELY collaborative* & the LH&W NCS (or CONVALESCENCE) Collaborative*

## Abstract

**Background** Biological evidence suggests ursodeoxycholic acid (UDCA)—a common treatment of cholestatic liver disease—may prevent severe COVID-19 outcomes. We aimed to compare the hazard of COVID-19 hospitalisation or death between UDCA users versus non-users in a population with primary biliary cholangitis (PBC) or primary sclerosing cholangitis (PSC). **Methods** With the approval of NHS England, we conducted a population-based cohort study using primary care records between 1 March 2020 and 31 December 2022, linked to death registration data and hospital records through the OpenSAFELY-TPP platform. Cox proportional hazards regression was used to estimate hazard ratios (HR) and 95% confidence intervals (CI) for the association between time-varying UDCA exposure and COVID-19 related hospitalisation or death, stratified by geographical region and considering models unadjusted and fully adjusted for pre-specified confounders. **Results** We identify 11,305 eligible individuals, 640 were hospitalised or died with COVID-19 during follow-up, 400 (63%) events among UDCA users. After confounder adjustment, UDCA is associated with a 21% relative reduction in the hazard of COVID-19 hospitalisation or death (HR 0.79, 95% CI 0.67–0.93), consistent with an absolute risk reduction of 1.35% (95% CI 1.07%–1.69%). **Conclusions** We found evidence that UDCA is associated with a lower hazard of COVID-19 related hospitalisation and death, support calls for clinical trials investigating UDCA as a preventative measure for severe COVID-19 outcomes.

## Plain language summary

Ursodeoxycholic acid is a drug used to treat liver disease. It has been proposed that it may prevent severe COVID-19 outcomes, however previous studies of this have had inconsistent results. We used electronic health records from people in the UK and identified people with two liver diseases: primary biliary cholangitis and primary sclerosing cholangitis. We looked at differences in hospitalisation and death between people taking UDCA and people who were not taking it. We found UDCA reduced the risk of severe COVID-19 outcomes by one-fifth. This suggests UDCA may help prevent serious COVID-19. Further clinical studies of UCDA should be undertaken, particularly in other groups with high risk or hospitalisation and death from COVID.

Ursodeoxycholic acid (UDCA) is first-line therapy in the treatment of primary biliary cholangitis (PBC) and commonly prescribed for people with primary sclerosing cholangitis (PSC). Both PBC and PSC are comparatively rare liver diseases that can lead to cirrhosis and end-stage liver disease. PBC

typically affects females more than males and most commonly presents at 50–60 years of age[1,2], while PSC is more common in men and presents at younger ages, often with comorbid inflammatory bowel disease[3,4]. UDCA has been shown to delay the progression of PBC, is usually prescribed

[1]London School of Hygiene and Tropical Medicine, London, UK. [2]Collaborative Centre for Organ Donation Evidence, Sydney School of Public Health, Faculty of Medicine and Health, University of Sydney, Camperdown, NSW, Australia. [3]NHMRC Clinical Trials Centre, University of Sydney, Camperdown, NSW, Australia. [4]Department of Gastroenterology & Hepatology, Concord Repatriation General Hospital, Concord, NSW, Australia. [5]Cambridge Liver Unit, Cambridge University Hospitals NHS Foundation Trust, Cambridge, UK. [6]Academic Department of Medical Genetics, University of Cambridge, Cambridge, UK. [7]Bennett Institute for Applied Data Science, Nuffield Department of Primary Care Health Sciences, University of Oxford, Oxford, UK. [8]Department of Internal Medicine, Yale School of Medicine, New Haven, CT, USA. [9]These authors jointly supervised this work: John Tazare, Christopher T Rentsch. *Lists of authors and their affiliations appear at the end of the paper. ✉e-mail: ruth.costello@lshtm.ac.uk

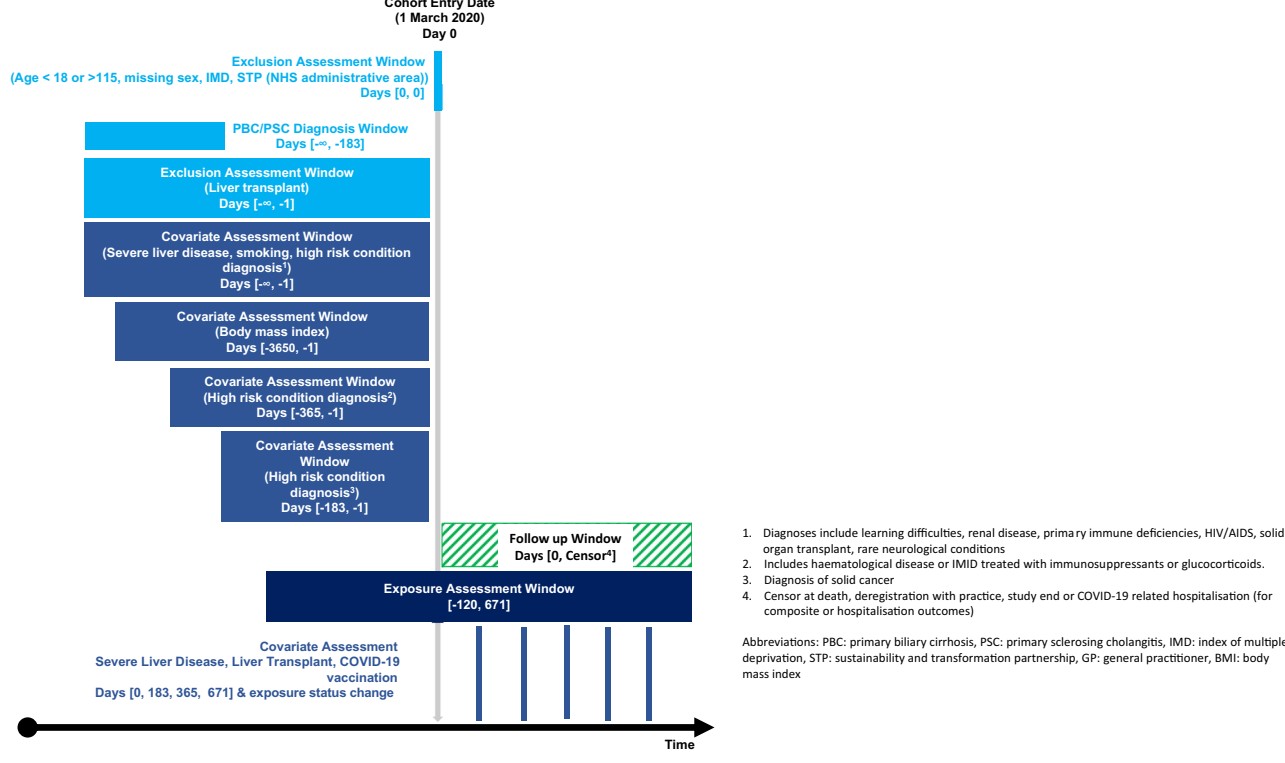

**Fig. 1 | Study design.**

lifelong[1,5], and is generally well-tolerated. A UK audit found that <10% of patients with PBC discontinued UDCA, with nausea, diarrhoea and vomiting the most frequent intolerances[6].

There is some biological evidence suggesting that UDCA protects against SARS-CoV-2 infection[7–9]. The proposed mechanism is that UDCA suppresses the signalling of the farnesoid X receptor (FXR) which reduces expression of angiotensin-converting enzyme 2 (ACE2), a cellular membrane protein which is the main receptor of SARS-CoV-2. Reduced ACE2 limits opportunities for SARS-CoV-2 to enter cells and reduces viral replication after infection. Whilst cohort and registry studies have investigated the association between UDCA exposure and severe COVID-19 outcomes, conflicting results were obtained[7,10–15]. These studies had important limitations relating to sample size, small numbers of outcomes and UDCA exposure measurement at a single time point or based on exposure during the study period. In addition, the extent to which vaccination has modified this association is unclear. The value of clinical trials of UDCA as prophylaxis or treatment for COVID-19, despite strong biological plausibility of benefit, in an era of high vaccination coverage is therefore unclear.

To address this evidence gap, we used routinely-collected data covering 43% of the population in England[16], to estimate the hazard of the composite outcome of COVID-19 related hospitalisation or death between 1 March 2020 and 31 December 2022, comparing use of UDCA treatment versus no UDCA treatment among people with PBC or PSC. We show that UDCA treatment is associated with a 21% lower hazard of COVID-19 related hospitalisation and death compared to no treatment.

## Methods
### Study design
We conducted a population-based cohort study using primary care records managed by the GP software provider TPP, linked to Office of National Statistics (ONS) death registration data and National Health Service (NHS) Secondary Use Service (SUS) data through OpenSAFELY, a data analytics platform created by our team on behalf of NHS England to address urgent COVID-19 research questions (https://opensafely.org).

**Data source.** All data were linked, stored and analysed securely using the OpenSAFELY platform, https://www.opensafely.org/, as part of the NHS England OpenSAFELY COVID-19 service. Data include pseudonymised data such as coded diagnoses, medications and physiological parameters. No free text data were included.

**Study population.** The study population included people with either PBC or PSC, defined as the presence of a SNOMED CT (diagnosis) code for PBC or PSC ≥6 months before index date (1st March 2020), and therefore indicated for treatment with UDCA. Since PBC and PSC are comparatively rare, we included both conditions to increase the power of the study. People were excluded if they had: (1) <18 or >115 years of age, (2) <6 months of registration in a TPP practice at the index date, which could preclude adequate ascertainment of key covariates, (3) missing information on sex, sustainability and transformation partnership (STP) region (an NHS administrative geographical area) or index of multiple deprivation (IMD), likely indicative of poor data quality or (4) a liver transplant prior to the index date, identified using SNOMED-CT codes in the primary care records or OPCS codes in secondary care records. People were followed from the index date until the earliest of either death, deregistration from their GP or the end of the study period (31st December 2022) (Fig. 1).

### Study measures
**Exposure.** The primary exposure was time-varying UDCA status. Baseline exposure status was defined as having at least one UDCA prescription during the 120 days prior to 1st March 2020. Prescriptions during follow-up were additionally identified and exposure status, including switches from unexposed to exposed and vice versa were updated accordingly. Given only the availability of prescription start dates we assumed treatment duration to be 120 days from prescription start date in primary analyses (derived from an assumed prescription

length of up to 90 days, and a gap of up to 30-days between prescription refills). If two prescriptions overlapped, the 120-days exposure period restarted on the later prescription; overlapping days were not added to the exposure time in primary analyses. This was informed by a descriptive analysis of the prescribing data and discussions with clinicians.

**Outcomes.** The primary outcome was a composite of COVID-19 related death or hospitalisation. The secondary outcomes considered COVID-19 related death and COVID-19 related hospitalisation individually. Deaths were identified using linked ONS death registration data. COVID-19 related death was defined as a death where the underlying or contributory cause on the death certificate was COVID-19 (ICD-10 codes U07.1 and U07.2). COVID-19 related hospitalisations, obtained from secondary care SUS data, were defined as any hospitalisation listing a COVID-19 diagnosis in any position (ICD-10 codes U07.1 and U07.2). For the composite outcome, if a person was hospitalised before death, the date of hospitalisation was used.

**Covariates.** Covariates were identified through literature review and discussions with domain experts. We extracted the following fixed covariates at index date: age, sex, ethnicity, deprivation, STP region (an NHS administrative geographical area), body mass index, smoking status, and presence of a COVID-19 high-risk diagnosis (i.e., learning difficulties, solid cancer, haematological disease, stem cell transplant, renal disease, immune-mediated inflammatory disorders identified through immunosuppressant drugs and glucocorticoid prescribing, primary immune deficiencies, HIV/AIDS, solid organ transplant, or rare neurological conditions including multiple sclerosis, motor neurone disease, myasthenia gravis or huntington's disease). COVID-19 high-risk diagnoses were identified through SNOMED CT codes in primary care records and ICD-10 and OPCS codes in secondary care records, and people without a code were assumed not to have a diagnosis. Ethnicity was identified through SNOMED CT codes in primary care records and supplemented with information from secondary care records. Recorded ethnicity was grouped into White versus non-White due to small numbers of people and outcomes in non-White ethnicities. Deprivation was measured using quintiles of the IMD, a relative measure of deprivation based on a person's postcode[17]. Body mass index was ascertained from weight measurements within the 10 years prior to index date, restricted to those taken when the patient was aged 16 years or older. Smoking status (never/former/current) was defined by the most recent SNOMED-CT code prior to the index date. Exposure to obeticholic acid (OCA), a second line therapy used in PBC, at index date was identified from linked high cost drug data[18]. This contains prescribing from April 2018 to March 2020, a person was considered exposed if they had at least one prescription between 1st September 2019 and 31st March 2020.

Time-varying covariates were assessed every six months and at date of exposure switching[19]. Time-varying covariates included: COVID-19 vaccination, liver transplant, and liver disease severity. COVID-19 vaccinations were identified using SNOMED-CT codes; people were considered vaccinated from the date of their first vaccination. Liver transplants were identified using the same codelist as for the exclusion criteria (described above). Liver disease severity was identified using SNOMED-CT codes and ICD-10 codes, which included decompensation events including hepatic encephalopathy, ascites and variceal haemorrhage. A person was assumed to have severe disease after the date of the earliest code.

**Missing data**
On the assumption that both obesity and smoking are more likely to be recorded if present, people with missing body mass index were assumed to be within healthy range, and people with missing smoking information were assumed never smokers, in line with previous work[20]. A missing category was used for people with missing ethnicity information.

**Statistical analysis**
The study population was described in a flowchart. The characteristics of the population were summarised using descriptive statistics, stratified by UDCA exposure status at index date and disease population.

Incidence rates by UDCA exposure status were calculated. Cox proportional hazards models, stratified by STP region, were used to estimate hazard ratios (HR) and 95% confidence intervals (CI) for the association between UDCA exposure and each outcome considering unadjusted, age and sex adjusted and fully adjusted models. Robust standard errors were applied.

In sensitivity analyses, we first added overlapping prescription days to exposure time. Second, we varied the assumed duration of UDCA prescriptions to be 90 days. Third, we analysed each population (i.e., PBC or PSC) separately. Fourth, we excluded people with OCA prescriptions at index date. OCA increases FXR levels, as UDCA decreases FXR levels, OCA may potentially have opposing effects on SARS-CoV-2 infection[7]. Fifth, to explore the impact of missing data we conducted a complete case analysis, including only those with non-missing data for smoking, BMI and ethnicity.

In secondary analyses, we aimed to estimate treatment effects in a population with high vaccination coverage. We repeated the primary analysis assuming an index date of 1st March 2021, as by this date a large majority of adults aged 60 years and older in the UK had received at least one COVID-19 vaccination[21]. In this analysis, we extended the adjustment for vaccination status by considering (a) days since the most recent vaccination at index date, categorised into quintiles and (b) a time-varying count of COVID-19 vaccinations over follow-up, this was reassessed, every 6-months and at date of exposure switching.

We derived cumulative incidence plots using Royston–Parmer models, standardised to the covariate distribution in the exposed groups, with the baseline hazard modelled using a spline with 2 degrees-of-freedom.

All counts are rounded to the nearest five to minimise potential disclosure. Data management was performed using Python 3.8, with analysis carried out using Stata 17. Code for data management and analysis as well as codelists available online (https://github.com/opensafely/udca_covid). All iterations of the pre-specified study protocol are archived (https://github.com/opensafely/udca_covid/docs).

**Information governance and ethical approval**
NHS England is the data controller of the NHS England OpenSAFELY COVID-19 Service; TPP is the data processor; all study authors using OpenSAFELY have the approval of NHS England[22]. This implementation of OpenSAFELY is hosted within the TPP environment which is accredited to the ISO 27001 information security standard and is NHS IG Toolkit compliant[23]. Further details are in supplementary note 1. This study was approved by the Health Research Authority (REC reference 20/LO/0651) and by the LSHTM Ethics Board (reference 21863). Data was collected under notices initially issued in February 2020 by the the Secretary of State under Regulation 3(4) of the Health Service (Control of Patient Information) Regulations 2002 (COPI Regulations), which required organisations to process confidential patient information for COVID-19 purposes; this set aside the requirement for patient consent.

**Reporting summary**
Further information on research design is available in the Nature Portfolio Reporting Summary linked to this article.

**Results**
We identified 11,305 people with PBC or PSC who met the inclusion criteria (supplementary Fig. 1). Of those, 7225 (64%) were exposed to UDCA at index date. Those prescribed UDCA at baseline were slightly older (aged 61–80 years: 55% UDCA vs 45% no UDCA), were more likely to be female (81% UDCA vs 74% no UDCA), and had more severe liver disease at baseline (42% UDCA vs 25% no UDCA) (Table 1). Characteristics stratified by disease population are also provided in supplementary Tables 1 and 2.

**Table 1 | Characteristics of the cohort by ursodeoxycholic acid (UDCA) exposure status at index date**

| Characteristic | | Overall N = 11,305 n (%) | No UDCA at index date N = 4080 n (%) | UDCA at index date N = 7225 n (%) |
|---|---|---|---|---|
| Population | PBC | 8800 (77.9) | 2780 (68.2) | 6020 (83.3) |
| | PSC | 2505 (22.1) | 1300 (31.8) | 1205 (16.7) |
| Age category | 18–40 years | 905 (8) | 505 (12.3) | 400 (5.5) |
| | 41–60 years | 2935 (26) | 1090 (26.7) | 1845 (25.5) |
| | 61–80 years | 5850 (51.7) | 1845 (45.3) | 4005 (55.4) |
| | >80 years | 1620 (14.3) | 640 (15.7) | 980 (13.6) |
| Sex | Female | 8855 (78.3) | 3035 (74.4) | 5820 (80.6) |
| | Male | 2450 (21.7) | 1045 (25.6) | 1405 (19.4) |
| Index of multiple deprivation | 1 (Most deprived) | 2080 (18.4) | 805 (19.8) | 1275 (17.6) |
| | 2 | 2035 (18) | 725 (17.8) | 1310 (18.1) |
| | 3 | 2475 (21.9) | 890 (21.8) | 1590 (22) |
| | 4 | 2410 (21.3) | 850 (20.8) | 1560 (21.6) |
| | 5 (Least deprived) | 2305 (20.4) | 810 (19.8) | 1500 (20.7) |
| Ethnicity | White | 10600 (93.8) | 3740 (91.7) | 6860 (94.9) |
| | Asian | 60 (0.5) | 20 (0.4) | 40 (0.6) |
| | Black | 380 (3.4) | 185 (4.5) | 195 (2.7) |
| | Mixed | 100 (0.9) | 65 (1.5) | 35 (0.5) |
| | Other | 90 (0.8) | 35 (0.9) | 55 (0.8) |
| | Unknown | 75 (0.7) | 35 (0.9) | 40 (0.5) |
| Severe liver disease | No | 7290 (64.5) | 3070 (75.2) | 4220 (58.4) |
| | Yes | 4015 (35.5) | 1010 (24.8) | 3005 (41.6) |
| Smoking status | Never | 4235 (37.5) | 1665 (40.9) | 2570 (35.5) |
| | Former | 1460 (12.9) | 570 (14) | 885 (12.3) |
| | Current | 5575 (49.3) | 1820 (44.6) | 3755 (52) |
| | Unknown[b] | 35 (0.3) | 20 (0.5) | 15 (0.2) |
| Body mass index[a] | Underweight | 305 (2.7) | 155 (3.8) | 155 (2.1) |
| | Healthy range | 3505 (31) | 1300 (31.9) | 2205 (30.5) |
| | Overweight | 3590 (31.7) | 1195 (29.3) | 2390 (33.1) |
| | Obese | 2720 (24) | 960 (23.5) | 1760 (24.4) |
| | Severe obesity | 425 (3.8) | 155 (3.8) | 270 (3.8) |
| | Unknown | 760 (6.7) | 315 (7.8) | 445 (6.1) |
| COVID high risk condition | No | 9615 (85.1) | 3465 (84.9) | 6150 (85.1) |
| | Yes | 1690 (14.9) | 615 (15.1) | 1075 (14.9) |
| Prescribed obeticholic acid at baseline | No | 11200 (99.1) | 4065 (99.6) | 7135 (98.7) |
| | Yes | 105 (0.9) | 15 (0.4) | 90 (1.3) |
| UDCA status switched at any point during follow-up | | 2715 (24.0) | 725 (17.8) | 1990 (27.5) |

[a]Body mass index categories: <18.5: underweight, 18.5–24.9: healthy range, 25–29.9: overweight:, 30–39.9: obese: ≥40, severe obesity. Those with unknown BMI were categorised into "healthy range" in the regression models.

[b]Those with unknown smoking status were categorised as "never" in the regression models.

There were 640 (5.7%) events of COVID-19 related hospitalisations or deaths during 29,834 person-years of follow-up. Individually there were 140 (1.2%) COVID-19 related deaths and 610 (5.4%) COVID-19 related hospitalisations. There were lower rates of COVID-19 related hospitalisation or death in those receiving UDCA users compared to non-users, with a rate of 176 events per 100,000 person-months in UDCA users, versus 184 events per 100,000 person-months in non-users. A similar pattern was seen for the individual outcomes of COVID-19 related hospitalisation and COVID-19 related death (Table 2).

In unadjusted analysis, there was no evidence of an association between UDCA exposure and the composite outcome (HR 0.91, 95% CI: 0.78–1.07) (Table 2 and Fig. 2). After full adjustment UDCA was associated with a 21% reduction in the hazard of COVID-19 related hospitalisation or death (HR 0.79, 95% CI: 0.67–0.93). When analysing COVID-19 hospitalisations and deaths separately, the fully adjusted HRs were 0.81 (95% CI: 0.68–0.96) and 0.76 (95% CI: 0.53–1.08), respectively. For all analyses, the validity of the proportional assumption was assessed through investigation of Schoenfeld residual plots and there were no observed violations (supplementary Figs. 2–4).

The standardised cumulative incidence for COVID-19 related hospitalisations or deaths was 6.1% (95% CI: 5.2%–7.2%) among UDCA users and 7.4% (95% CI: 6.2% –8.8%) among non-users (Fig. 3). The absolute cumulative risk difference was −1.35% (95% CI: −1.07% to −1.69%).

In sensitivity analyses, results were robust after adding overlapping days to exposure time (HR 0.81, 95% CI: 0.68–0.95), considering a maximum prescription duration of 90 days (HR 0.79, 95% CI: 0.68–0.93), excluding people prescribed OCA (HR 0.80, 95% CI: 0.68–0.94), and exclusion of people with missing information for smoking, BMI or ethnicity (HR: 0.79, 95% CI: 0.67–0.94). When modelling populations separately, we found associations were primarily driven by the PBC population, likely due to relatively few events in the PSC population (Fig. 2 and supplementary Tables 3–8).

In the secondary analysis, there were 11,550 people who met the inclusion criteria as of 1st March 2021. There were 45 COVID-19 related deaths and 410 COVID-19 related hospitalisations, combined to a total of 420 COVID-19 related hospitalisations or deaths. The crude HR was 0.83 (95% CI: 0.68–1.01) and after adjustment the HR was 0.71 (95% CI: 0.58–0.87), in line with the main analysis (Fig. 2 and supplementary Table 9).

## Discussion

In this large cohort study of people with PBC or PSC, use of UDCA was associated with a 21% lower hazard of COVID-19 related hospitalisation and death. Our results were consistent with a maximum absolute risk reduction of 1.7% in the context of an absolute risk of 7.4% of COVID-19 related hospitalisation or death among non-users. Our findings were robust to a variety of sensitivity analyses. Our data covered the start of the pandemic, emerging variants, and time before and after most adults were fully vaccinated. We further explored the association in a population where the majority had at least one vaccination, as most of the population is now vaccinated. In a secondary analysis limited to a study period in which a large majority of adults were vaccinated, we showed that UDCA was associated with 29% reduction in COVID-19 related hospitalisation or death.

There have been six observational studies of varying quality[10–15] using routinely-collected and survey data investigating the potential benefits of UDCA exposure to prevent and treat COVID-19. Two studies have shown that UDCA exposure was associated with reduced COVID-19 outcomes, including SARS-CoV-2 infection, COVID-19 hospitalisation, and death due to COVID-19[7,10,11]. The largest study was a US cohort of 1607 male adults with cirrhosis that found a 48% reduction in severe or critical COVID-19 associated with UDCA use[10]. However, the study population included a variety of liver diseases that may have different susceptibilities to COVID-19, which the authors acknowledge may not have been adequately adjusted for[10,11]. Further, UDCA status was based on any use during follow-up, which may have resulted in exposure misclassification. Our study limited the study population to PSC and PBC to allow for better confounding control, and UDCA exposure was time-updated during follow-up.

Three studies showed no evidence of an association between UDCA and COVID-19 related or all-cause death. These were conducted in

**Table 2 | Rates and Cox proportional hazard model results for each outcome[a]**

| Outcome | Exposure status | Number of events | Rate per 100,000 person-months | Unadjusted model hazard ratio (95% confidence interval) | Age and sex adjusted model hazard ratio (95% confidence interval) | Fully adjusted model hazard ratio (95% confidence interval)[b] |
|---|---|---|---|---|---|---|
| Composite (hospitalisation or death) | No UDCA | 240 | 184.0 | Reference | Reference | Reference |
| | UDCA | 400 | 175.8 | 0.91 (0.78–1.07) | 0.88 (0.75–1.04) | 0.79 (0.67–0.93) |
| COVID-19 related death | No UDCA | 55 | 41.3 | Reference | Reference | Reference |
| | UDCA | 85 | 36.9 | 0.82 (0.58–1.15) | 0.86 (0.61–1.21) | 0.76 (0.53–1.08) |
| COVID-19 related hospitalisation | No UDCA | 225 | 175.8 | Reference | Reference | Reference |
| | UDCA | 385 | 169.2 | 0.94 (0.79–1.10) | 0.9 (0.77–1.07) | 0.81 (0.68–0.96) |

[a]All models clustered by region (STP).

[b]Adjusted for age, sex, high risk conditions at baseline, ethnicity, indices of multiple deprivation, body mass index, severe liver disease (time-varying), COVID-19 vaccination (time-varying), liver transplant (time-varying).

**Fig. 2 | Forest plot of hazard ratio and 95% confidence intervals for UDCA vs no UDCA for each outcome in the main analysis and sensitivity analyses (*n* = 11,305).** Death alone outcome for PSC only sensitivity analysis not available due to small number of deaths.

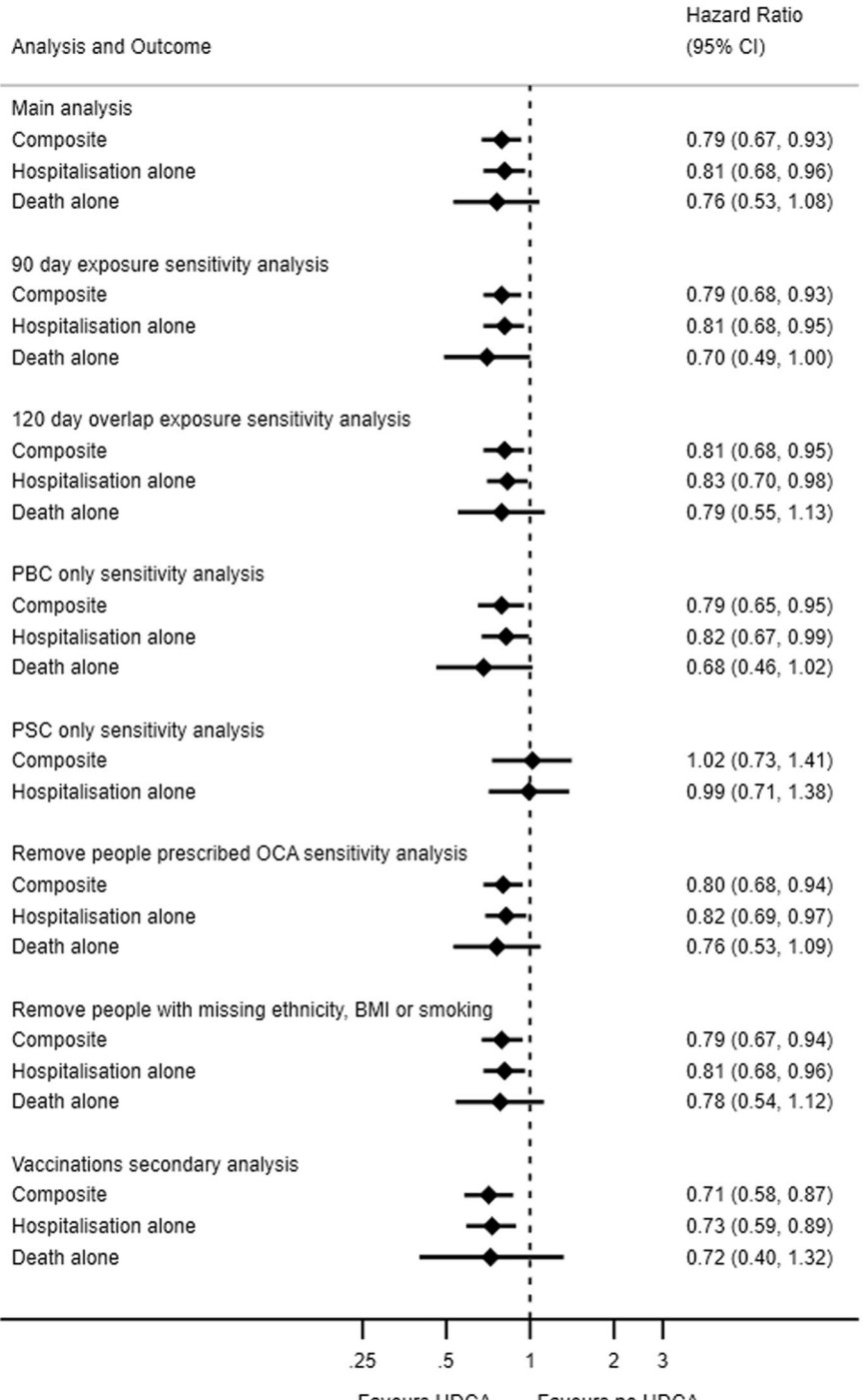

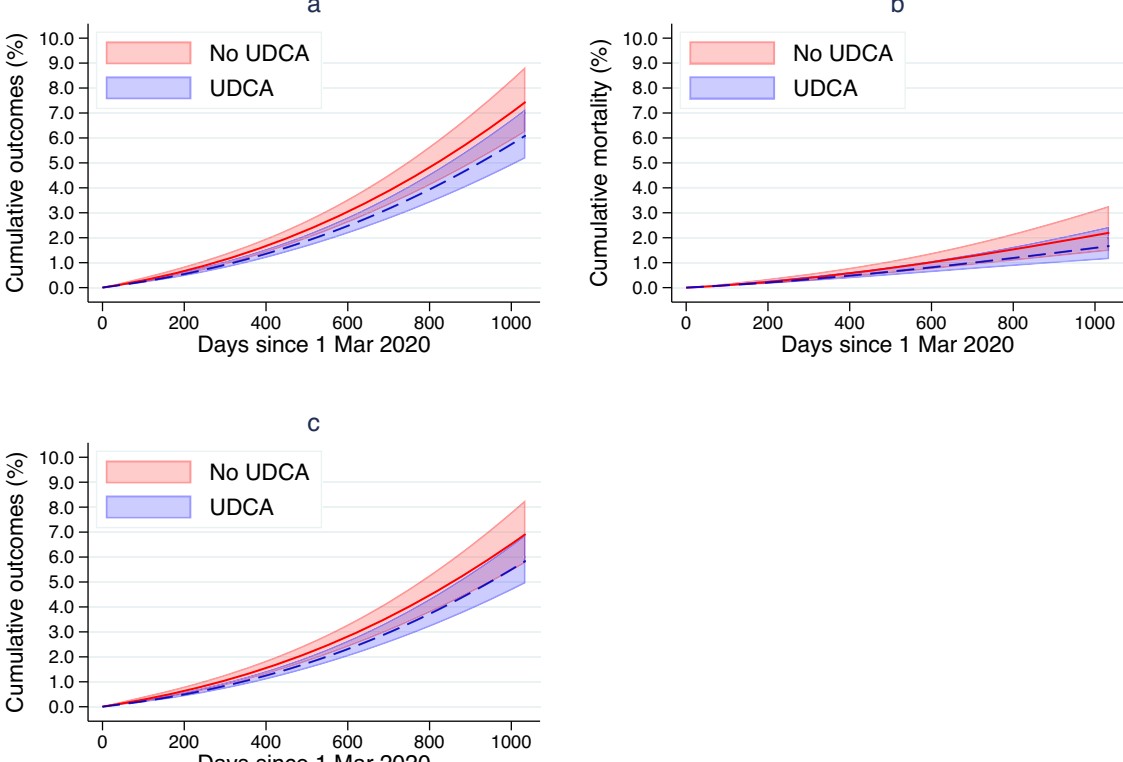

**Fig. 3 | Standardised cumulative incidence curves.** The panels show standardised cumulative incidence curves for (**a**) composite outcome of COVID-19 hospitalisation or death, (**b**) COVID-19 death only, (**c**) COVID-19 hospitalisation only. The red line refers to the cumulative incidence estimate for the no UDCA group, the dashed blue line refers to the cumulative incidence estimate for the UDCA group. Shaded areas refer to the 95% confidence interval of the cumulative incidence.

populations with established SARS-CoV-2 infection or hospitalised with COVID-19 and the numbers treated with UDCA were small[12–14]. Using a study population with acute SARS-CoV-2 infection focuses the study question on UDCA as treatment of COVID-19. The present study aimed to estimate the potential benefit of using UDCA as preventative therapy, though our design does not fully elucidate the mechanism that drives the benefit observed. Of note, two of these studies did observe a reduced need for continuous positive airway pressure[14] and a lower proportion with ICU admissions[13] in those treated with UDCA compared to no UDCA.

Strengths of this study include the use of a large database of around 28 million records[16] which allowed us to identify a study population of over 11,000 people. We used a specific study population of people with PBC or PSC, which allowed us to identify and address confounding specific to this population. We identified a comprehensive set of potential confounders, and adjusted for these in a time-varying manner where this was deemed important, such as liver disease severity. Our results were robust to sensitivity analyses, including varying the assumed prescription duration, although power was limited when restricting to the PSC population alone. There are also limitations to acknowledge. Firstly, duration of UDCA exposure was estimated as we only had the date of prescription available and we did not have hospital prescribing information. We would not expect any potential exposure misclassification to vary by outcome status; therefore, any bias would be towards the null. Second, prevalence of exposure at index was lower than expected[6], suggesting the potential for exposure misclassification. However, we implemented time-varying UDCA exposure, reducing misclassification of exposure status over time. Previous studies have either determined exposure at a single time point or based on 'ever' exposure during the study period, which increases the potential for exposure misclassification at the time of the outcome. We do not expect exposure misclassification to be differential by outcome status; therefore, any residual misclassification would bias our results towards the null. Third, we did not have information on UDCA dose so could not examine potential dose

thresholds or dose-response patterns. Fourth, as is the case in many nationwide electronic health record systems, we did not have reliable test data on SARS-CoV-2 infections covering the whole of the study period, and therefore could not examine this as an additional outcome. Fifth, we examined the potential for competing risk of liver-related death and found that, whilst there were more deaths with the primary cause as liver disease in the UDCA exposure group ($n = 175$, 2.4%) compared to the unexposed group ($n = 75$, 1.8%), the overall proportion experiencing liver-related death was small. In a post-hoc analysis, we removed censoring for liver-related deaths, which resulted in a slight attenuation in the estimated absolute cumulative risk difference (−1.35% versus −1.28%).

Although COVID-19 deaths and hospitalisations have substantially reduced since the height of the pandemic, there are still groups that remain at high risk of severe COVID-19 outcomes, despite vaccination[24]. While treatments have been introduced which reduce risk of severe COVID-19 outcomes after infection[25], vaccinations remain the only preventative measure. Therefore, further preventative measures are important for these groups. UDCA is a widely used, long-term treatment for people with liver disease that has a good safety profile[1] and is off-patent making it a good candidate for repurposing as a preventative measure for severe COVID-19 in high risk groups. Further to this, UDCA has few drug interactions, with 6 listed in the BNF[26], making it a good candidate for repurposing[27]. Our findings support basic science evidence that UDCA prevents severe COVID-19[7], though our data could not identify underlying mechanisms. Clinical trials have been called for[7,28], and may help to elucidate the mechanism of action UDCA as a potential preventative measure for severe COVID-19. To our knowledge, only two interventional clinical trials have been registered, one is a single-arm study in healthcare workers investigating COVID-19 infection (ClinicalTrials.gov ID: NCT05659654), the other is in children with COVID-19 infection investigating COVID-19 outcomes (ChiCTR ID: ChiCTR2200067226), both are short-term studies, and results have not yet been published.

Among patients with PBC and PSC, UDCA was associated with clinically-meaningful lower absolute risks and relative hazards of COVID-19 related hospitalisation and death, providing strong evidence that UDCA should be investigated further in observational and interventional studies, particularly in other high risk groups.

## Data availability

Access to the underlying identifiable and potentially re-identifiable pseudonymised electronic health record data is tightly governed by various legislative and regulatory frameworks, and restricted by best practice. The data in the NHS England OpenSAFELY COVID-19 service is drawn from General Practice data across England where TPP is the data processor. TPP developers initiate an automated process to create pseudonymised records in the core OpenSAFELY database, which are copies of key structured data tables in the identifiable records. These pseudonymised records are linked onto key external data resources that have also been pseudonymised via SHA-512 one-way hashing of NHS numbers using a shared salt. University of Oxford, Bennett Institute for Applied Data Science developers and PIs, who hold contracts with NHS England, have access to the OpenSAFELY pseudonymised data tables to develop the OpenSAFELY tools. These tools in turn enable researchers with OpenSAFELY data access agreements to write and execute code for data management and data analysis without direct access to the underlying raw pseudonymised patient data, and to review the outputs of this code. All code for the full data management pipeline—from raw data to completed results for this analysis—and for the OpenSAFELY platform as a whole is available for review at github.com/OpenSAFELY. The data management and analysis code for this paper was led by Ruth Costello and contributed to by John Tazare and Christopher Rentsch.

## Code availability

All code and codelists are shared openly for review and re-use under MIT open licence (https://github.com/opensafely/udca_covid). Detailed pseudonymised patient data is potentially re-identifiable and therefore not shared.

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

## Acknowledgements

We are very grateful for all the support received from the TPP Technical Operations team throughout this work, and for generous assistance from the information governance and database teams at NHS England and the NHS England Transformation Directorate. North East Commissioning Support Unit provided support on behalf of all Commissioning Support Units to

aggregate the high cost drugs data for use in OpenSAFELY studies. We thank Dr Edward Parker for his help with output checking. The OpenSAFELY Platform is supported by grants from the Wellcome Trust (222097/Z/20/Z) and MRC (MR/V015737/1, MC_PC_20059, MR/W016729/1). In addition, development of OpenSAFELY has been funded by the Longitudinal Health and Wellbeing strand of the National Core Studies programme (MC_PC_20030: MC_PC_20059), the NIHR funded CONVALESCENCE programme (COV-LT-0009), NIHR (NIHR135559, COV-LT2-0073), and the Data and Connectivity National Core Study funded by UK Research and Innovation (MC_PC_20058) and Health Data Research UK (HDRUK2021.000). L.A.T. is funded by an NIHR Research Professorship NIHR302405. V.M. is funded by an NIHR Doctoral Research Fellowship (NIHR301535).

## Author contributions

L.A.T. conceived the idea. R.E.C., J.T., C.T.R., L.A.T., K.M.J.W., R.S., G.F.M., A.W. and A.S. were involved in the development of the study. R.E.C., J.T., C.T.R., E.H., B.Z., L.Y.L. and V.M. had access to the data. R.E.C., J.T. and C.T.R. were responsible for data management and statistical analysis. R.E.C. wrote the first draft of the manuscript. All authors contributed to and approved the final manuscript, and accept responsibility to submit for publication.

## Competing interests

B.G. has received research funding from the Bennett Foundation, the Laura and John Arnold Foundation, the NHS National Institute for Health Research (NIHR), the NIHR School of Primary Care Research, NHS England, the NIHR Oxford Biomedical Research Centre, the Mohn–Westlake Foundation, NIHR Applied Research Collaboration Oxford and Thames Valley, the Wellcome Trust, the Good Thinking Foundation, Health Data Research UK, the Health Foundation, the World Health Organisation, UKRI MRC, Asthma UK, the British Lung Foundation, and the Longitudinal Health and Wellbeing strand of the National Core Studies programme; he is a Non-Executive Director at NHS Digital; he also receives personal income from speaking and writing for lay audiences on the misuse of science. BMK is also employed by NHS England working on medicines policy and clinical lead for primary care medicines data. A.M. is a member of RCGP health informatics group and the NHS Digital GP data Professional Advisory Group, and received consulting fee from Induction Healthcare. L.A.T. has received research funding from MRC, Wellcome, NIHR and GSK, consulted for Bayer in relation to an observational study of chronic kidney disease (unpaid), and is a member of 4 non-industry funded (NIHR/MRC) trial advisory committees (unpaid) and MHRA Expert advisory group (Women's Health). R.E.C. has shares in AstraZeneca. V.M. received a grant from NIHR. A.S. is employed by LSHTM on a fellowship sponsored by GSK. J.T. received an AstraZeneca grant for unrelated COVID-19 research. G.F.M. and R.S. have received research funding from Intercept Pharmaceuticals and Advanz Pharma. All other authors declare no conflicts of interest.

## Additional information

**Peer review information** Ursodeoxycholic acid and severe COVID-19 outcomes in a cohort study using the OpenSAFELY platform.

## the OpenSAFELY collaborative

Ruth E. Costello [1] ✉, Angel Y. S. Wong [1], Anna Schultze[1], Viyaasan Mahalingasivam[1], Emily Herrett [1], Bang Zheng[1], Liang-Yu Lin[1], Brian MacKenna[7], Amir Mehrkar [7], Sebastian C. J. Bacon[7], Ben Goldacre[7], Laurie A. Tomlinson [1], John Tazare[1,9] & Christopher T. Rentsch [1,8,9]

## the LH&W NCS (or CONVALESCENCE) Collaborative

Ruth E. Costello [1] ✉, Viyaasan Mahalingasivam[1], Emily Herrett [1], Bang Zheng[1], Brian MacKenna[7], Ben Goldacre[7], Laurie A. Tomlinson [1] & John Tazare[1,9]

A full list of members and their affiliations appears in the Supplementary Information.

