## [Peer Review File · Communications Medicine]

Reviewers' comments:

Reviewer #1 (Remarks to the Author):

Thank you for inviting me to review this engaging manuscript. The paper presents timely and compelling evidence. The research examines the link between ursodeoxycholic acid and the severity of COVID-19 outcomes in individuals with liver disease, employing a cohort study via the OpenSAFELY platform. The manuscript is effectively structured, with both the abstract and discussion clearly articulating the core message. The sections detailing data sources and linkages, exposure measures, outcomes, covariates, and handling of missing data are thorough and lucid. The statistical techniques applied are apt, and the findings are succinctly communicated. Furthermore, the graphical and tabular representations are of commendable quality, although I recommend some minor adjustments.

On page 7, regarding missing data, it would be beneficial for the authors to explain the handling or rationale behind the absence of data for other covariates, such as conditions deemed high-risk for COVID-19, and their strategies for managing such gaps.

On the same page, the authors' approach of introducing a 'missing' category for individuals with unspecified ethnicity within regression analysis is a common yet statistically debatable method. This 'missing-indicator' method is criticized for potentially skewing parameter estimates, even when data is missing completely at random. Alternatives, albeit not without their own limitations, include omitting incomplete risk factors or employing listwise deletion under the assumption of missing data being completely random. Advanced techniques such as multiple imputation or Full Information Maximum Likelihood (FIML) estimation, although preferable, may pose challenges due to the extensive dataset.

In Table 1, given the assertion that individuals with unrecorded smoking status were considered non-smokers, I propose integrating the 'unknown' category with 'Never smokers'.

For Figure 3, adding a footnote to clarify the meaning of the shaded area would enhance understanding.

It would also be pertinent to discuss the potential implications of drug interactions, particularly the interplay between various clinical medications and the severity of COVID-19, further enriching the study's context and relevance, as supported by literature from

<https://jamanetwork.com/journals/jamanetworkopen/article-abstract/2791291>

[https://www.thelancet.com/journals/eclinm/article/PIIS2589-5370\(23\)00241-9/fulltext](https://www.thelancet.com/journals/eclinm/article/PIIS2589-5370(23)00241-9/fulltext)

<https://www.sciencedirect.com/science/article/pii/S075333222100425X>

Reviewer #2 (Remarks to the Author):

The authors investigate a potential protective role of UDCA to prevent severe COVID-19 outcomes. The study is generally well-designed and studies a large amount of patients with these rare diseases. Statistical approach is sound and references are up to date. The results are not completely novel, but have an important added value to the current literature on UDCA and COVID.

I have a few concerns/remarks.

- The fact that no hospital prescription information is available (to stratify the patients into UDCA and no UDCA) is a major concern. According to table 1 around one third of PBC patients is not treated with UDCA. It seems extremely unlikely that that proportion of patients with PBC is not treated with UDCA. UDCA is the cornerstone of PBC treatment and has almost no contra-indications. Furthermore, like the authors state in their introduction, UDCA is rarely stopped by PBC patients due to its limited side effects. This makes the results difficult to interpret since patients solely followed in a hospital setting have probably another clinical profile (e.g. more decompensated liver disease compared to recompensated).

- How do the authors explain the difference between PBC and PSC patients? Are there clinical differences (e.g. younger age) that “protect” PSC patients from developing COVID?

- Please add the clinical information (comparable to table 1) for PBC and PSC patients separately (at least as supplemental information).

- The authors suggest that UDCA could be used as a treatment for severe COVID-infections. Would this be feasible due to the large NNT (absolute risk reduction 1%)?

Re. (COMMSMED-24-0043-T) Ursodeoxycholic acid and severe COVID-19 outcomes in people with liver disease: a cohort study using the OpenSAFELY platform

We are delighted that Communications medicine is considering publishing our manuscript. We are grateful for the time and attention of the external reviewers and editorial team. We have made substantial changes to the manuscript based on these comments and feel their input has greatly strengthened the manuscript. Please find below the reviewer comments (in italics) followed by our responses. Note that page numbers refer to locations in the clean version of the revised manuscript.

Reviewer #1 (Remarks to the Author):

1.1 Thank you for inviting me to review this engaging manuscript. The paper presents timely and compelling evidence. The research examines the link between ursodeoxycholic acid and the severity of COVID-19 outcomes in individuals with liver disease, employing a cohort study via the OpenSAFELY platform. The manuscript is effectively structured, with both the abstract and discussion clearly articulating the core message. The sections detailing data sources and linkages, exposure measures, outcomes, covariates, and handling of missing data are thorough and lucid. The statistical techniques applied are apt, and the findings are succinctly communicated. Furthermore, the graphical and tabular representations are of commendable quality, although I recommend some minor adjustments.

On page 7, regarding missing data, it would be beneficial for the authors to explain the handling or rationale behind the absence of data for other covariates, such as conditions deemed high-risk for COVID-19, and their strategies for managing such gaps.

Response: Thank you for your time to review our manuscript and provide constructive feedback. We assumed that people without a primary care (SNOMED-CT) or hospital code (ICD-10 or OPCS) for conditions deemed at high risk for COVID-19 did not have that condition. Therefore, there was no missing data for this variable. This has been clarified in the covariates section (page 6) by adding “, and people without a code were assumed not to have a diagnosis” Variables with missing values are described in the missing data section on page 7. As these are serious conditions, we expect them to be coded in the primary care or secondary care records if they were present, resulting in minimal misclassification.

1.2 On the same page, the authors' approach of introducing a 'missing' category for individuals with unspecified ethnicity within regression analysis is a common yet statistically debatable method. This 'missing-indicator' method is criticized for potentially skewing parameter estimates, even when data is missing completely at random. Alternatives, albeit not without their own limitations, include omitting incomplete risk factors or employing listwise deletion under the assumption of missing data being completely random. Advanced techniques such as multiple imputation or Full Information Maximum Likelihood (FIML) estimation, although preferable, may pose challenges due to the extensive dataset.

Response: Thank you for allowing us to expand on our rationale for dealing with missing data. Recent work has clarified the assumptions under which using a missing category approach is appropriate [1]. Informally, this approach is valid when a variable only acts as confounder when observed; i.e. it is not used in the treatment decision when unobserved. When using electronic health record data, we believe this assumption is likely to be at least approximately true. We were concerned that multiple imputation would not be suitable for missing ethnicity, smoking and body mass index as these were potentially in violation of the assumption for missing at random. We

agree there are limitations with using the missing category for ethnicity, in particular, and therefore have included a complete case analysis as an additional sensitivity analysis (see Methods section, page 7). Results from the main analyses were robust to this new sensitivity analysis and estimates have been added to Figure 2, page 14, and supplementary materials.

- [1] Blake HA, Leyrat C, Mansfield KE, Tomlinson LA, Carpenter J, Williamson EJ. Estimating treatment effects with partially observed covariates using outcome regression with missing indicators. *Biom J* 2020;62:428–43. <https://doi.org/10.1002/bimj.201900041>.

1.3 In Table 1, given the assertion that individuals with unrecorded smoking status were considered non-smokers, I propose integrating the 'unknown' category with 'Never smokers'.

Response: We reported the unknown smoking status and BMI in Table 1 for transparency of the distribution of missingness by exposure category. We have clarified in updated footnotes to Table 1 (page 11) that we categorised those with unknown BMI and unknown smoking status into “healthy range” and “never”, respectively, in the regression models.

1.4 For Figure 3, adding a footnote to clarify the meaning of the shaded area would enhance understanding.

Response: As suggested, a footnote has been added (Results section, page 16) as follows:

“The red line refers to the cumulative incidence estimate for the no UDCA group, the dashed blue line refers to the cumulative incidence estimate for the UDCA group. Shaded areas refer to the 95% confidence interval of the cumulative incidence.”

1.5 It would also be pertinent to discuss the potential implications of drug interactions, particularly the interplay between various clinical medications and the severity of COVID-19, further enriching the study's context and relevance, as supported by literature from

<https://jamanetwork.com/journals/jamanetworkopen/article-abstract/2791291>

[https://www.thelancet.com/journals/eclinm/article/PIIS2589-5370\(23\)00241-9/fulltext](https://www.thelancet.com/journals/eclinm/article/PIIS2589-5370(23)00241-9/fulltext)

<https://www.sciencedirect.com/science/article/pii/S075333222100425X>

Response: Thank you for this important suggestion. We have added the following sentence to the discussion (page 18) to emphasise the potential utility and safety of UDCA: “Further to this, UDCA has few drug interactions, with 6 listed in the BNF [26], making it a good candidate for repurposing [27].”

[26] Ursodeoxycholic acid | Interactions | BNF content published by NICE n.d.

<https://bnf.nice.org.uk/interactions/ursodeoxycholic-acid/> (accessed April 24, 2024).

[27] Conti V, Sellitto C, Torsiello M, Manzo V, De Bellis E, Stefanelli B, et al. Identification of Drug Interaction Adverse Events in Patients With COVID-19: A Systematic Review. *JAMA Netw Open* 2022;5:e227970.

Reviewer #2 (Remarks to the Author):

2.1 The authors investigate a potential protective role of UDCA to prevent severe COVID-19 outcomes. The study is generally well-designed and studies a large amount of patients with these rare diseases. Statistical approach is sound and references are up to date. The results are not completely novel, but have an important added value to the current literature on UDCA and COVID.

I have a few concerns/remarks. The fact that no hospital prescription information is available (to stratify the patients into UDCA and no UDCA) is a major concern. According to table 1 around one third of PBC patients is not treated with UDCA. It seems extremely unlikely that that proportion of patients with PBC is not treated with UDCA. UDCA is the cornerstone of PBC treatment and has almost no contra-indications. Furthermore, like the authors state in their introduction, UDCA is rarely stopped by PBC patients due to its limited side effects. This makes the results difficult to interpret since patients solely followed in a hospital setting have probably another clinical profile (e.g. more decompensated liver disease compared to recompensated).

Response: Thank you for your time to review our manuscript and provide constructive feedback. Although one third of the study cohort including individuals with PBC or PSC were not prescribed UDCA at index date (1st March 2020), 18% switched from unexposed to exposed during the study period (see Table 1, page 10). Therefore, 82% of the study cohort were exposed to UDCA during follow-up. The findings for the PBC cohort alone are similar (supplementary materials). These figures are in line with findings of an audit of 11 NHS hospitals in the UK that found 84.8% of patients had records of UDCA prescribing [2]. We discuss the lack of hospital prescribing as a limitation in the discussion (page 17). In the revised manuscript, we add that, “We would not expect any potential exposure misclassification to vary by outcome status; therefore, any bias would be toward the null.”.

- [2] Sivakumar M, Gandhi A, Shakweh E, Li YM, Safinia N, Smith BC, et al. Widespread gaps in the quality of care for primary biliary cholangitis in UK. *Frontline Gastroenterol* 2022;13:32–8. <https://doi.org/10.1136/flgastro-2020-101713>.

2.2 How do the authors explain the difference between PBC and PSC patients? Are there clinical differences (e.g. younger age) that “protect” PSC patients from developing COVID?

Response: Thanks for this comment. Our data do not suggest that PSC patients are differentially protected from COVID as the rates of COVID-19 hospitalisations and deaths were similar between the PBC and PSC groups. As expected, the PSC population was slightly different to the PBC population in that they were younger, more proportionately male, and had less severe liver disease but slightly more comorbid high risk COVID-19 conditions at index date (see pages 4-8 of the supplementary materials). These were all adjusted for in the multivariable analysis. We note that the confidence intervals for the treatment effect in the PSC group include the point estimates from the PBC group. We are therefore cautious to definitively conclude meaningful differences in the treatment effect between these groups. We also consistently showed evidence of lower hazard of COVID-19 severe outcomes in PBC patients only. Therefore the age composition of PSC patients was unlikely to affect the interpretation of the main findings.

2.3 Please add the clinical information (comparable to table 1) for PBC and PSC patients separately (at least as supplemental information).

Response: Thanks for this suggestion, we have added this table to the supplementary materials and the following text to the results (Page 8):

“Characteristics stratified by disease population are also provided in supplementary materials.”

2.4 The authors suggest that UDCA could be used as a treatment for severe COVID-infections. Would this be feasible due to the large NNT (absolute risk reduction 1%)?

Response: We believe that our work highlights the potential for UDCA as a preventative therapy rather than a treatment after diagnosis. For example, people at high risk of severe COVID-19 outcomes could be prescribed UDCA to prevent serious outcomes during a wave of a new variant where vaccine effect may be impaired. As UDCA is a low cost drug that is off-patent, has few side effects, and has only a few known drug-drug interactions (see Response 1.5 above), it would seem feasible to use UDCA as a preventative treatment in patients where COVID-19 could become severe. We call for more evidence from further interventional studies in people at high risk of COVID-19 to confirm efficacy and provide support for a positive benefit-risk ratio, including estimates of absolute risk reduction in these populations, which could be used to inform situations where this is considered cost-effective.

REVIEWERS' COMMENTS:

Reviewer #1 (Remarks to the Author):

I find that the revisions have been comprehensive and the manuscript is reasonable for publication

Reviewer #2 (Remarks to the Author):

The answers from the authors sufficiently addressed my concerns.